# Outcomes of Outpatient Elective Esophageal Varices Band Ligation in Cirrhosis Patients with Significant Thrombocytopenia

**DOI:** 10.3390/diseases13020027

**Published:** 2025-01-23

**Authors:** Nisar Amin, Mark Ayoub, Julton Tomanguillo, Harleen Chela, Veysel Tahan, Ebubekir Daglilar

**Affiliations:** 1Department of Internal Medicine, Charleston Area Medical Center, West Virginia University, Charleston, WV 25304, USA; mark.ayoub@camc.org; 2Division of Gastroenterology and Hepatology, Charleston Area Medical Center, West Virginia University, Charleston, WV 25304, USA; julton.tomanguillo@vandaliahealth.org (J.T.); harleen.chela@camc.org (H.C.); veysel.tahan@vandaliahealth.org (V.T.)

**Keywords:** cirrhosis, varices, band ligation, thrombocytopenia, mortality, transfusion

## Abstract

Background: Current guidelines advise against platelet transfusion prior to emergent esophageal variceal band ligation (EVL) in cirrhotic patients with platelet counts below 50 × 10^3^/μL. However, recommendations for elective EVL remain unclear. This study evaluates the outcomes of cirrhotic patients undergoing outpatient EVL. Methods: Adult patients aged 18 years and older diagnosed with cirrhosis, with or without significant thrombocytopenia (<50 × 10^3^/μL), were identified using the TriNetX database. Patients who received platelet transfusions within one week prior to or on the day of EVL were excluded. Cirrhotic patients with significant thrombocytopenia undergoing outpatient elective EVL were categorized into two cohorts: (1) those with platelet counts between 30 and 49 × 10^3^/μL and (2) those with platelet counts ≥50 × 10^3^/μL. Propensity score matching (PSM) was employed to compare rates of post-EVL esophageal variceal bleeding and 14-day mortality between the two cohorts. Results: A total of 16,718 cirrhotic patients undergoing outpatient EVL were included in the analysis. Of these, 17.2% (n = 2874) had significant thrombocytopenia, while 82.8% (n = 13,844) had platelet counts ≥50 × 10^3^/μL. Two well-matched cohorts (2864 patients each) were created using 1:1 PSM. No statistically significant differences were observed between the groups regarding 14-day post-EVL esophageal variceal bleeding (13.7% vs. 15.2%; *p* = 0.12), 14-day mortality (5.7% vs. 5.0%; *p* = 0.28), and 28-day mortality (8.4% vs. 7.5%; *p* = 0.20). Conclusions: Elective EVL appears to be safe in cirrhotic patients with platelet counts as low as 30 × 10^3^/μL, challenging the current threshold of 50 × 10^3^/μL for platelet transfusion.

## 1. Introduction

Esophageal and gastric variceal hemorrhages are severe complications of chronic liver disease (CLD), significantly contributing to morbidity and mortality. Variceal bleeding occurs in up to 40% of patients with cirrhosis, with each episode carrying a mortality rate of approximately 20% [1,2]. Recent data indicate a 135% increase in hospitalizations due to esophageal varices over the past decade [3], underscoring the need for effective management strategies to improve patient outcomes.

For the prophylaxis of esophageal variceal bleeding, non-selective beta-blockers (NSBBs) are generally considered the first-line treatment. Endoscopic variceal band ligation (EVL) serves as an alternative for patients with moderate to severe varices, either in combination with NSBBs or as an option for those who are intolerant of, or have contraindications to, NSBBs [4]. Despite its efficacy, EVL is associated with bleeding complications, occurring in 2–10% of patients with CLD. These complications may arise during the procedure or 1–2 weeks post-ligation due to ulcer formation [5,6].

Clinical evidence suggests that a high Child–Pugh score, elevated model for end-stage liver disease (MELD) score, and the presence of ascites are significant risk factors for post-EVL ulcer bleeding [7,8]. Additionally, reflux esophagitis has been identified as a factor increasing the risk of bleeding in patients undergoing EVL [4]. While bleeding-risk assessment is critical for clinical decision-making in patients with cirrhosis undergoing EVL, no established criteria exist for accurately predicting bleeding risk. Traditional coagulation biomarkers, such as INR, PT, and PTT, are considered unreliable for patients with CLD due to the complex nature of hemostasis in this population [9].

Thrombocytopenia is the most common hematologic abnormality observed in CLD patients. Although platelet count is often used as a bleeding-risk predictor, its reliability remains a topic of debate. Current expert opinion recommends a pre-procedure platelet count cutoff of 50 × 10^3^/μL; however, emerging studies question this threshold [7,8,10]. Notably, research indicates that post-EVL bleeding is not consistently associated with baseline platelet counts, and platelet transfusions might paradoxically increase the bleeding risk by increasing portal hypertension from excessive volume expansion [7].

Balancing the need for blood product transfusion with safety concerns is critical, as platelet transfusions, while effective in preventing or controlling bleeding in thrombocytopenic patients, carry inherent risks, such as transfusion reactions, alloimmunization, and infections. This multinational, multicenter, large-scale study seeks to evaluate the safety of performing esophageal variceal band ligation (EVL) in patients with severe thrombocytopenia, with platelet counts as low as 30 × 10^3^/μL.

## 2. Materials and Methods

This retrospective cohort study utilized longitudinal medical data obtained from the Global Collaborative Network, which contains 120 healthcare organizations of TriNetX Research Network. TriNetX is an extensive electronic medical records database containing de-identified information for over 270 million patients across 120 healthcare organizations throughout 30 countries. The data available in this database are comprehensive and include patient demographics, diagnoses, procedures, medications, laboratory tests, and healthcare utilization. The anonymized nature of the data ensures patient privacy, while the extensive data available within the platform provide researchers with a wealth of information to analyze.

We queried the TrinetX Research Network between 2006 and 2024. We included patients above the age of 18 who have a diagnosis of cirrhosis and esophageal varices, are undergoing outpatient esophagogastroduodenoscopy, and have platelets ≥30 × 10^3^/μL. We excluded patients aged over 90 years and those who received platelet transfusions either on the day of the procedure or within one week prior to ligation. Lists of the ICD-10 and the Current Procedural Terminology (CPT) codes used to identify patients in this study are shown in Table 1.

Two cohorts were created for the study. The first cohort included adult patients with cirrhosis who had significant thrombocytopenia, defined by platelets between 30 and 49 × 10^3^/μL. The second cohort included adult patients with cirrhosis who had platelets ≥50 × 10^3^/μL. To reduce the impact of confounding factors, a propensity score analysis was used in order to create groups with matched baseline characteristics and comorbidities. A study flow diagram is shown in Figure 1.

TriNetX’s built-in function was used to match the two groups at a 1:1 ratio by nearest neighbor, matching for age at index, hypertension, diabetes mellitus, chronic kidney disease, chronic obstructive pulmonary disease (COPD), coronary artery disease (CAD), malignancy, heart failure, ascites, SBP, PPI use, Child–Pugh, and MELD variables. In all the analyses, statistical significance was considered when a 95% confidence interval (95% CI) and *p*-value < 0.05 were observed. A Kaplan–Meier method was used for the survival probability analysis. The study’s primary outcome was the incidence of esophageal variceal bleeding within 14 days post-band-ligation. The secondary outcome was the mortality rate within the same timeframe. The secondary outcome was the mortality rate at 14 and 28 days. A subgroup analysis was performed with platelet threshold of 20 × 10^3^/μL.

## 3. Results

A total of 16,718 cirrhotic patients who underwent outpatient elective band ligation were included in this analysis. Of these, 17.2% (n = 2874) had significant thrombocytopenia, while 82.8% (n = 13,844) had platelet counts ≥50 × 10^3^/μL. Two well-matched cohorts (2864 patients each) were created using 1:1 PSM. Prior to PSM, patients with significant thrombocytopenia had a lower age when compared to patients with platelets ≥50 × 10^3^/μL (54.0 ± 12.6 vs. 57.3 ± 12.5, *p* < 0.01). A comorbidity analysis of both cohorts revealed that patients with significant thrombocytopenia had a higher prevalence of CKD (22% vs. 16%), COPD (13% vs. 11%), ascites (64% vs. 55%), SBP (14% vs. 8%), hepatic encephalopathy (19% vs. 12%), ESRD (6% vs. 4%), and hemodialysis (4% vs. 2%). Conversely, patients with platelets ≥50 × 10^3^/μL had higher rates of hypertension (55% vs. 53%) and diabetes mellitus (DM) (39% vs. 37%). Both cohorts had similar rates of atherosclerotic heart disease (16% vs. 17%), heart failure (14% vs. 13%), and neoplasms (41% vs. 41%). Proton pump inhibitor use was significantly higher in the low-platelet group (82% vs. 73%). Additionally, mean values of bilirubin and INR were higher in the low-platelet group, whereas creatinine, sodium, and albumin levels were comparable between the two groups.

Propensity score matching (PSM) was performed using baseline patient characteristics and comorbidities, including demographics, comorbid conditions, proton pump inhibitor (PPI) use, and variables from the MELD and Child–Pugh scores. After PSM, there was no statistically significant difference between the first cohort, consisting of patients with significant thrombocytopenia, and the second cohort, which included patients with platelets ≥50 × 10^3^/μL. A full comparison of the two cohorts before and after PSM is presented in Table 2.

Two well-matched cohorts were established using a 1:1 propensity-score-matching model. Each cohort, post-matching, included 2864 patients. The post-matched analysis demonstrated that patients with significant thrombocytopenia (platelet count of 30–49 × 10^3^/μL) did not exhibit a significantly higher rate of esophageal variceal bleeding within 14 days following band ligation (13.7% vs. 15.2%, *p* = 0.26), with an odds ratio (OR) of 0.90 and a 95% CI of (0.79, 1.03). Furthermore, overall mortality was not significantly different between the two groups at 14 days (5.7% vs. 5.0%, *p* = 0.24) and 28 days (8.4% vs. 7.5%, *p* = 0.20) post-band-ligation. The outcomes are summarized in Table 3, with a corresponding bar graph presented in Figure 2. Similarly, a subgroup analysis comparing platelet thresholds of 20 × 10^3^/μL and 50 × 10^3^/μL revealed no significant differences in 14-day esophageal variceal bleeding rates (13.9% vs. 15.0%; *p* = 0.22) or mortality rates (5.5% vs. 4.7%; *p* = 0.16) between the groups. Additionally, the post-endoscopic variceal ligation (EVL) endoscopy rate was comparable between the two groups at the two-week mark (7.5% vs. 8.3%; *p* = 0.23).

## 4. Discussion

### 4.1. Overall Summary of Our Study

This study investigates the role of platelet count as a predictor of bleeding in cirrhotic patients undergoing prophylactic endoscopic variceal band ligation (EVL). To the best of our knowledge, this is the largest study to date evaluating the efficacy of platelet transfusion in preventing post-EVL bleeding in this patient population. The analysis includes data from over 16,000 patients across 120 healthcare organizations in 30 countries who underwent the procedure.

Our findings indicate no significant difference in the incidence of esophageal variceal bleeding within 14 days post-procedure (13.7% vs. 15.2%, *p* = 0.12) or in mortality (57% vs. 5.2%, *p* = 0.24) when comparing platelet thresholds of 30 × 10^3^/μL and 50 × 10^3^/μL. A subgroup analysis comparing platelet thresholds of 20 × 10^3^/μL and 50 × 10^3^/μL revealed no significant differences in 14-day esophageal variceal bleeding (14% vs. 15%; *p* = 0.22) or mortality (5.5% vs. 4.7%; *p* = 0.16) between the two groups. These results align with previously reported outcomes associated with platelet levels below 50 × 10^3^/μL.

Notably, this study is the first to directly compare post-procedure bleeding and all-cause mortality in cirrhotic patients undergoing EVL using a platelet threshold as low as 20 × 10^3^/μL.

### 4.2. Hemostasis

Hemostasis is a complex process in cirrhosis, as chronic liver disease significantly alters the production of hemostatic factors by the liver. Due to the balanced impact on both procoagulant and anticoagulant pathways, patients typically reach a new hemostatic equilibrium. As a result, traditional bleeding markers, such as PT, PTT, and INR, are not reliable predictors of bleeding in this patient population. Whole-blood viscoelastic tests, including thromboelastography (TEG) and rotational thromboelastography (ROTEM), are primarily used to guide transfusion decisions in cases of acute bleeding during invasive procedures in patients with cirrhosis. However, these tests are not reliable for predicting bleeding in elective procedures such as prophylactic EVL. Thrombocytopenia, with a reported prevalence of 64–84%, is the most common hematologic abnormality observed in patients with chronic liver disease (CLD) [11,12]. The causes of thrombocytopenia in CLD are multifactorial and include blood pooling in the enlarged spleen due to portal hypertension, hypersplenism, decreased platelet production, and bone marrow suppression. Establishing an accurate and reliable cutoff for platelet transfusion is critical in cirrhotic patients due to the risks associated with blood product transfusions, including infections, transfusion reactions, and volume overload. Furthermore, unnecessary transfusion-induced volume expansion may exacerbate portal hypertension, consequently increasing the risk of variceal bleeding.

### 4.3. Cirrhosis Screening

Cirrhosis can result in portal hypertension, a condition in which portal pressure increases due to blood flow impedance caused by the architectural distortion of the liver as a result of ongoing fibrosis [13]. Additionally, intrahepatic vasoconstriction, triggered by alteration in endogenous nitric oxide, further contributes to elevated portal pressure [14,15]. This increased pressure leads to the development of collateral vessels, including esophageal varices. Esophageal varices are the most clinically significant complication of portal hypertension due to their high mortality risk. They are present in approximately 50% of patients with cirrhosis, and their presence correlates with the severity of the disease [16]. The gold standard for diagnosing varices is esophagogastroduodenoscopy (EGD). According to the guidelines from the American Association for the Study of Liver Diseases (AASLD), it is recommended that patients with cirrhosis undergo EGD screening for varices at the time of diagnosis [17,18]. The recommended screening intervals depend on the size and severity of the esophageal varices.

### 4.4. EVL Safety

EVL is the preferred method for endoscopic intervention in cases of acute variceal bleeding and for preventing variceal rebleeding [19,20]. While complications of EVL are not uncommon, they are generally minor. The most frequent complications include transient dysphagia and chest discomfort [21]. Ulcer formation at the site of ligation may result in minor bleeding [22]. However, more severe complications can occur, such as incomplete ligation or insufficient vessel occlusion, which may lead to intraoperative bleeding, death, or early or late rebleeding. Typically, the EVL ring remains attached to the esophageal wall for up to seven days, but premature dislodgment can occur. This premature detachment exposes the original bleeding vessel, significantly increasing the risk of rebleeding. Additional complications, such as abdominal pain and infection, have also been reported [21]. Variceal bleeding is more commonly observed in patients with worse liver function, primarily due to uncontrolled portal hypertension. This correlation is evident in patients with higher risks of early bleeding who have elevated MELD scores and Child–Pugh C status.

### 4.5. Predictors of Rebleed Post-EVL

Incidence of early post-EVL bleeding was 7.7% in a large meta-analysis of 16 studies. Our study findings are supported by a large systematic review in 2022 that was able to identify numerous factors that predict early rebleeding [23]. A total of 14 factors were found to be significantly associated with early rebleeding, as highlighted in Table 4.

None of them included platelets; however, coagulopathy was present at least three times, which platelets typically play major role in. Of note, their analysis was unable to find an association between the value of prothrombin time (PT), international normalized ratio (INR), or platelet count [23]. This further supports our study findings.

### 4.6. Platelets and Bleed

Hemostasis is a complex physiological process involving platelets, coagulation cascades, fibrinolytic systems, blood vessels, and cytokines. It is activated in response to tissue injury and can be divided into three distinct phases. Primary hemostasis encompasses blood vessel vasoconstriction, and the formation of a platelet plug through platelet adhesion and aggregation. Secondary hemostasis involves the activation of the coagulation cascade, leading to the deposition and stabilization of fibrin. Finally, tertiary hemostasis focuses on the dissolution of the fibrin clot through plasminogen activation to prevent excessive clotting [24,25].

Platelets play a critical role in hemostasis by initiating the process that prevents bleeding. Upon vascular injury, platelets adhere to the exposed sub-endothelial matrix, become activated, and aggregate to form a primary hemostatic plug, which is essential for halting initial blood loss [26].

Thrombocytopenia, or low platelet count, significantly increases the risk of bleeding due to the insufficient formation of an effective hemostatic plug. This risk is particularly concerning in the context of invasive procedures. Non-cirrhotic patients with platelet counts below 50,000/μL are at heightened risk of bleeding, with the risk escalating markedly when platelet counts fall below 10–15 × 10⁹/L. The clinical context, as well as the specific procedure or condition, also plays a crucial role in determining bleeding risk [27].

### 4.7. Cirrhosis, Platelets, and Bleeding

Patients with cirrhosis have a very complex coagulation system that puts them at risk for both bleeding and clotting, which makes traditional means to monitor for bleeding or clotting, such as PT/INR or platelets, unreliable [28]. Furthermore, a decompensating event, such as variceal bleed, is often preceded by an inciting event such as infection, which independently may have an effect on the coagulation system in those patients [29]. Additionally, the fibrinolytic system that regulates blood clot formation, remodeling, and breakdown is affected in both pro- and anti-fibrinolytic pathway [28]. Such an effect causes an imbalance which can be tilted towards thrombosis or accelerated fibrinolysis. One of the mechanisms through which the coagulation pathway is affected is that patients with cirrhosis have variable degrees of thrombocytopenia, which is due to increased platelet destruction, increased splenic and hepatic sequestration, and decreased levels of thrombopoietin. These patients also have impaired thromboxane A2 production and abnormal GPIb, which also adds to platelet dysfunction [30,31,32,33]. Since patients with cirrhosis have an impaired coagulation and bleeding pathway and often have concomitant thrombocytopenia, it is crucial to evaluate the bleeding risk in those patients who have thrombocytopenia [28]. This prompted us to evaluate the risk of bleeding in patients with cirrhosis who have significant thrombocytopenia after elective variceal banding.

### 4.8. Recommendations

There is significant discrepancy between the recommendations of major GI societies regarding platelet transfusion in patients with cirrhosis [34]. The American Gastroenterology Association (AGA) and European Association for the Study of the Liver (EASL) both advise against using FFP for PT/INR correction [35,36]. However, the AGA, in their guidelines, stated that we may use platelet transfusion to target a goal 50 × 10^3^ u/L. That being said, this recommendation was based on a low level of evidence [35]. Other societies, including Baveno VII, remain neutral in their recommendations due to lack of significant evidence [37].

### 4.9. Transfusion

Current literature shows that the transfusion of FFP or platelets may increase the procoagulant factor levels, thrombin, and platelet count in stable patients; however, it remains unknown if such transfusions provide any clinical benefit [38]. The evidence that supports platelet transfusion for thrombocytopenia correction is drawn from studies evaluating prophylactic platelet transfusion in patients with cirrhosis to minimize post-operative bleeding related to elective procedures [39,40,41]. A multicenter analysis by Blasi et al., which evaluated the role of prophylactic transfusion of platelets in patients with cirrhosis who are undergoing EVL, found that the incidence of post-EVL bleeding was low and was not related to the platelet count. In fact, most patients with post-EVL bleeding did not meet the platelet threshold of 50 × 10^3^ /μL that warrants prophylactic transfusion [7]. This also supports the findings of our study, which show that at a lower cutoff of 30 × 10^3^ u/L, platelet transfusion did not provide any benefit. Similarly, Biswas et al., when comparing patients who did not receive platelet transfusion to those who did, found that patients receiving transfusions were more likely to have a decompensating event, a higher post-EVL rebleeding rate, and a higher mortality rate [34].

### 4.10. Our Study

Our study demonstrates no significant difference in post-procedure bleeding and all-cause mortality in patients with a lower platelet cutoff of 30 × 10^3^ /μL when compared to the currently accepted cut-off of 50 × 10^3^ /μL for elective variceal band ligation in those with chronic liver disease. This finding is important and clinically quite relevant, as it has the potential to alter the approach to variceal band ligation. Endoscopic variceal band ligation is considered to be a high-risk procedure with the potential for precipitating further hemorrhage; hence, gastroenterologists tend to be rather cautious in this scenario. They may forego elective band ligation when the platelet count is below 50,000/μL. However, with the findings derived from this study, perhaps a less timid approach may be considered with less need for platelet transfusions and avoiding potential delays of care as well and not to mention the risks associated with transfusion of blood products themselves. Utilizing lower cut off values for platelet counts as a parameter to safely perform endoscopy in cirrhotic patients and treat varices may prove to be revolutionary. As outlined above, the coagulation cascade and hemostatic pathways are significantly altered in cirrhotic patients. This study highlights that we can be more liberal in our approach towards band ligation and still be able to carry it out safely without the need for pre-procedure transfusion of platelets. Therefore, we propose adopting an individualized approach for patients with cirrhosis undergoing invasive procedures. Furthermore, we recommend reevaluating the currently employed platelet cutoff of 50 × 10^3^/μL and a more cautious and conservative approach to the utilization of platelets in patients with cirrhosis who are undergoing prophylactic esophageal variceal band ligation.

### 4.11. Limitations and Strengths

This study represents the largest investigation to date on this topic, encompassing a substantial cohort of patients from multiple centers across various countries over a span of more than a decade. However, several limitations must be acknowledged. As a retrospective review, this study is inherently subject to weaknesses such as potential selection bias and confounding factors. Additionally, the de-identified nature of the retrospective database precludes consideration of the performing gastroenterologist’s expertise or objective findings during endoscopy.

Despite its limitations, this study has several notable strengths. The inclusion criteria were carefully designed to minimize confounding factors, thereby enhancing the validity of the findings. Additionally, the utilization of data from multiple countries and diverse healthcare systems increases the generalizability of the results. The application of propensity score matching further strengthens the study by reducing the influence of confounding variables. Although the database does not permit the direct calculation of MELD and Child–Pugh scores, variables within these classifications were accounted for, enabling the creation of two highly comparable cohorts. These methodological considerations underscore the reliability and robustness of the study’s conclusions.

## 5. Conclusions

Our multi-country study found no significant association between significant thrombocytopenia as low as 20 × 10^3^ /μL and 14-day post-procedural esophageal bleed in patients with cirrhosis who underwent outpatient elective variceal banding. There was also no significant association in terms of 14-day mortality post-procedure. This study enforces a universal implementation of the new cut-off, which is crucial to significantly reducing unnecessary platelet transfusions.

## Figures and Tables

**Figure 1 diseases-13-00027-f001:**
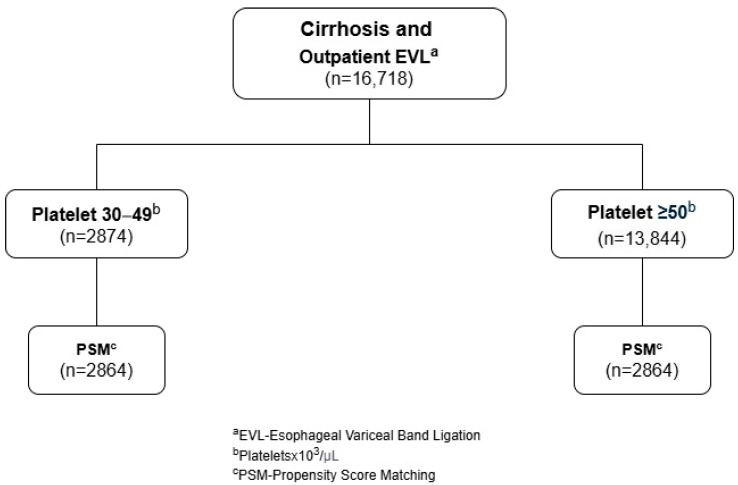
Study flow diagram.

**Figure 2 diseases-13-00027-f002:**
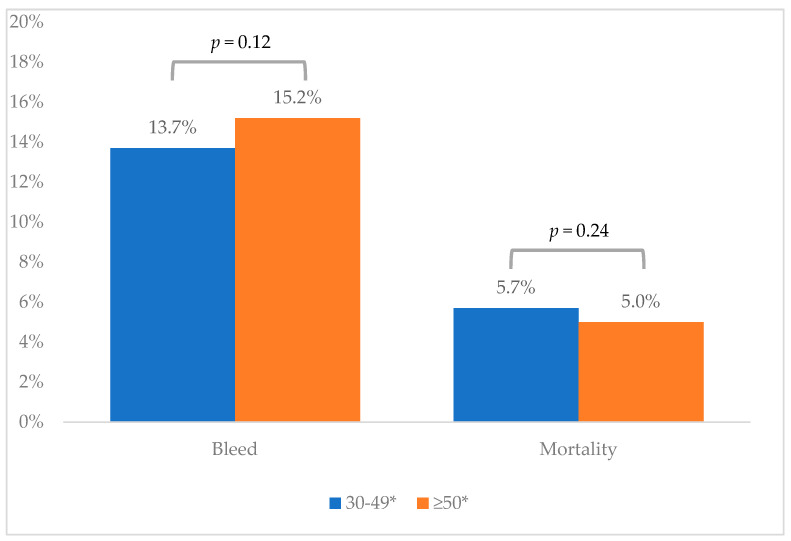
Bar graph of outcomes. * ×10^3^/μL.

**Table 1 diseases-13-00027-t001:** List of international classification of diseases—10 and current procedural terminology used in the study.

Diagnosis or Procedure	ICD-10 ^a^ or CPT ^b^
Esophageal Varices	I85.0
Cirrhosis	K70.2, K71.50, K72.1, K74.0, K74.1, K74.2, K74.6
Esophagogastroduodenoscopy	43,244
Outpatient Service	1,013,626

^a^ ICD—international classification of diseases—10; ^b^ CPT—current procedural terminology.

**Table 2 diseases-13-00027-t002:** Baseline characteristics of study cohorts before and after propensity matching.

	Before Propensity Score Matching			After Propensity Score Matching				
	Platelet Count			Platelet Count				
	30–49 × 10^3^/µL (n = 2874)	≥50 × 10^3^/µL (n = 13,844)	*p*-Value	30–49 × 10^3^ (n = 2864)	≥50 × 10^3^ (n = 2864)		*p*-Value	
Demographics								
Age at Index, Mean years ± SD	54.0 ± 12.6	57.3 ± 12.5			0.001	54.0 ± 12.6	53.9 ± 12.6	0.809
Female	915 (32)	4800 (41)			0.003	915 (32)	911 (32)	0.910
Male	1844 (64)	8450 (61)			0. 003	1843 (64)	1846 (65)	0.934
Race								
White	2087 (723)	10,196 (74)			0.184			0.678
Black or African American	198 (7)	961 (7)			0. 897	198 (7)	208 (7)	0.607
Comorbidities/medication								
Diabetes mellitus	1066 (37)	5399 (39)			0.046	1066 (37)	1039 (36)	0.459
Chronic kidney disease	632 (22)	2190 (16)			0.001	631 (22)	598 (21)	0.288
Chronic obstructive pulmonary disease	369 (13)	1551 (11)			0.014	369 (13)	379 (13)	0.695
Atherosclerotic heart disease	449 (16)	2297 (17)			0.186	449 (16)	457 (16)	0.772
Heart failure	391 (14)	1745 (13)			0.156	391 (14)	393 (14)	0.939
Essential (primary) hypertension	1515 (53)	7603 (55)			0.022	1514 (53)	1535 (54)	0.578
Neoplasms	1169 (41)	5679 (41)			0.664	1169 (41)	1147 (40)	0.554
End stage renal disease	182 (6)	476 (4)			0.001	181 (6)	173 (6)	0.661
Hemodialysis	103 (4)	251 (2)			0.001	102 (4)	115 (4)	0.368
Spontaneous bacterial peritonitis	390 (14)	1091 (8)			0.001	389 (14)	384 (14)	0.847
Hepatic encephalopathy	529 (19)	1705 (12)			0.001	528 (18)	496 (17)	0.270
Proton pump inhibitors	2348(82)	10,112 (73)			0.001	2347 (82)	2391 (84)	0.124
Lab Results								
Creatinine	1.1 ± 1.1	1.0 ± 1.1			0.001	1.1 ± 1.1	1.2 ± 1.1	0.66
Bilirubin Total	4.3 ± 6.6	2.7 ± 5.6			0.001	4.3 ± 6.5	4.0 ± 6.4	0.12
INR in Plasma or Blood	1.6 ± 0.6	1.4 ± 0.5			0.001	1.6 ± 0.6	1.6 ± 0.6	0.01
Sodium	137 ± 4.8	136 ± 4.6			0.11	137 ± 4.8	136 ± 4.9	0.87
Albumin	3.1 ± 0.7	3.2 ± 0.7			0.001	3.1 ± 0.7	3.1 ± 0.7	0.79

Values are mean ± SD or n (%) unless otherwise specified.

**Table 3 diseases-13-00027-t003:** Outcomes.

	14 Days			
	Platelets 30–49 * (n = 2864)	Platelets ≥50 * (n = 2864)	*p*-Value	OR (95% CI)
Bleed	13.7%	15.2%	0.12	0.90 (0.785, 1.03)
Mortality	5.7%	5.0%	0.24	

* ×10^3^/μL

**Table 4 diseases-13-00027-t004:** List of predictors of post-EVL bleeding.

Clinical Characteristics	Endoscopic Findings	Findings on Imaging	Laboratory Investigations	Scores
Age	Peptic esophagitis	Portal vein diameter	Hemoglobin	Child Pugh C status
Male sex	Grade III/large esophageal varices	Dilated portal vein	Serum bilirubin	MELD score
Emergency indication for EVL	Presence of high-risk stigmata	Portal vein thrombosis	Prothrombin time	MELD-Na score
Prior history of variceal bleeding	Whole extent of esophageal varices	Presence of hepatocellular carcinoma	PT-INR	APRI score
Presence of moderate to gross ascites	Number of varices		Prothrombin concentration	
Presence of hepatic encephalopathy	Gastric varices		Albumin	
Spontaneous bacterial peritonitis	Portal hypertensive gastropathy			
Presence of infection	Number of bands			
Plasma transfusion				
Use of proton pump inhibitors after EVL				
Use of beta-blockers after EVL				
Esophageal banding				

## Data Availability

The legal and ethical restrictions under which the data were provided do not allow for the data to be made publicly available. The data we used for this paper were acquired from TriNetX (https://www.trinetx.com/) on 10 January 2025 Release and/or sharing of these data are not covered under our data use agreement with TriNetX. To gain access to the data, a request can be made to TriNetX (moc.xtenirt@nioj), but costs may be incurred, and a data sharing agreement would be necessary.

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
