# Peer review of "Outcomes of Outpatient Elective Esophageal Varices Band Ligation in Cirrhosis Patients with Significant Thrombocytopenia"

_diseases, 2025, doi:10.3390/diseases13020027_

Round 1
Reviewer 1 Report
Comments and Suggestions for Authors
Authors aimed to assess the outcomes of cirrhotic patients who underwent EVL as an outpatient.
They found that mo significant difference was noted between cirrhotic patients with and without significant thrombocytopenia in the rate of post-EVL esophageal variceal bleeding, mortality at 14 days and 1-month. The end-points were not statistically different in both group in terms of 14-days esophageal varices bleeding (10.54% vs 12.50%, P=0.26), 14-days mortality (2.86%% vs 3.31%%, P=0.63), and 1-month mortality (4.51% vs 6.17%, P=0.17).
This is an interesting paper. There are several minor comments.
1) Was prophylactic use of PPI done ?
2) Please, describe the rate of post-EVL ulcer bleeding.
3) Was it a routine to perform 2nd-look EGD after EVL ?
Author Response
Reviewer #1
Authors aimed to assess the outcomes of cirrhotic patients who underwent EVL as an outpatient.
They found that no significant difference was noted between cirrhotic patients with and without significant thrombocytopenia in the rate of post-EVL esophageal variceal bleeding, mortality at 14 days and 1-month. The end-points were not statistically different in both group in terms of 14-days esophageal varices bleeding (10.54% vs 12.50%, P=0.26), 14-days mortality (2.86%% vs 3.31%%, P=0.63), and 1-month mortality (4.51% vs 6.17%, P=0.17).
This is an interesting paper. There are several minor comments. Thank you for your comments. The responses are provided below.
1) Was prophylactic use of PPI done? We performed propensity score matching (PSM) between the groups to match for PPI use.
2) Please, describe the rate of post-EVL ulcer bleeding? The database was updated with the current dates, and the updated rates of post-EVL are now reflected in the paper.
3) Was it a routine to perform 2nd-look EGD after EVL? We looked for EGD within 14 days of EVL, the outcome is incorporated in the “results” section.
Reviewer 2 Report
Comments and Suggestions for Authors
Dear Authors,
The article is interesting, but it needs some improvements:
- The abstract is solid, but it would benefit from a more detailed justification of the study's significance. This change would enhance its clarity, depth, and impact.
- The introduction provides a clear overview of the clinical problem and rationale for the study. However, it could benefit from a more focused structure that transitions smoothly from the general topic (variceal hemorrhage in CLD) to the specific research question (EBL in thrombocytopenic patients). Consider reorganizing the content to group related ideas more cohesively. For instance:
1) Start with the burden and clinical impact of variceal hemorrhage.
2) Introduce EBL as a treatment option, its indications, and associated risks.
3) Discuss the limitations of current practices, focusing on thrombocytopenia and platelet transfusion. The mention of esophageal obstruction due to banding is interesting but it doesn't directly tie into the focus on thrombocytopenia or platelet transfusion. This could be omitted or reframed to better align with the main research question. More details from the literature could be provided, and the research rationale could be expanded. These points are critical to the study's rationale and should be emphasized more prominently. Consider rephrasing the aim to make it more concise and direct.
- Metod is well-detailed and provides a clear outline of the study design, data source, patient selection criteria, and analytical methods. Also the table 1 and the figure are well-detailed and well rappresented.
- The Results section provides comprehensive data analysis and demonstrates careful methodological execution. The table 2, 3 and the figure 2 are clear and well descriptions. However the results on PSM and baseline characteristics are repeated unnecessarily. For instance: "There was no statistically significant difference between the cohorts in all PSM components" is mentioned multiple times. Streamline this to avoid redundancy.
- Discussion is very detailed. However, consider rephrasing to emphasize the novelty of the study earlier in the paragraph for better engagement. For example: "This is the largest study to date evaluating platelet transfusion thresholds for post-EBL bleeding, analyzing over 7,000 patients globally". This section effectively explains the complexities of hemostasis in cirrhotic patients. However:
1) The section "Platelets and Bleeding" delves into overly detailed descriptions of platelet physiology. These could be simplified to avoid diverting attention from the primary results and clinical implications.
2) The limitations are candidly discussed, but the strengths could be more prominently emphasized.
3) Table 4 is clear and detailed.
4) Avoid repetition and simplify verbose sections to prioritize key takeaways.
-In the conclusions highlight the practical implications and potential benefits of adopting the new platelet cut-off more clearly.
I suggest this article of which I am co-author to expand line 33-38; I think it is interesting to do a framework on mortality and in what mortality class the disease under study is identified.: Golinelli D, Guarducci G, Sanna A, Lenzi J, Sanmarchi F, Fantini MP, Montomoli E, Nante N. Regional and sex inequalities of avoidable mortality in Italy: A time trend analysis. Public Health Pract (Oxf). 2023 Oct 26;6:100449. doi: 10.1016/j.puhip.2023.100449.
Author Response
Reviewer #2
Dear Authors,
The article is interesting, but it needs some improvements:
- The abstract is solid, but it would benefit from a more detailed justification of the study's significance. This change would enhance its clarity, depth, and impact. Thank you for the constructive suugestions. The suggested changes have been incorporated into the introduction.
- The introduction provides a clear overview of the clinical problem and rationale for the study. However, it could benefit from a more focused structure that transitions smoothly from the general topic (variceal hemorrhage in CLD) to the specific research question (EBL in thrombocytopenic patients). Consider reorganizing the content to group related ideas more cohesively. For instance:
1) Start with the burden and clinical impact of variceal hemorrhage. The suggested changes applied.
2) Introduce EBL as a treatment option, its indications, and associated risks. The suggested changes applied.
3) Discuss the limitations of current practices, focusing on thrombocytopenia and platelet transfusion. The mention of esophageal obstruction due to banding is interesting but it doesn't directly tie into the focus on thrombocytopenia or platelet transfusion. This could be omitted or reframed to better align with the main research question. More details from the literature could be provided, and the research rationale could be expanded. These points are critical to the study's rationale and should be emphasized more prominently. Consider rephrasing the aim to make it more concise and direct. The suggested changes applied.
- Metod is well-detailed and provides a clear outline of the study design, data source, patient selection criteria, and analytical methods. Also the table 1 and the figure are well-detailed and well rappresented. We appreciate your input.
- The Results section provides comprehensive data analysis and demonstrates careful methodological execution. The table 2, 3 and the figure 2 are clear and well descriptions. However the results on PSM and baseline characteristics are repeated unnecessarily. For instance: "There was no statistically significant difference between the cohorts in all PSM components" is mentioned multiple times. Streamline this to avoid redundancy. Suggested changes applied.
- Discussion is very detailed. However, consider rephrasing to emphasize the novelty of the study earlier in the paragraph for better engagement. For example: "This is the largest study to date evaluating platelet transfusion thresholds for post-EBL bleeding, analyzing over 7,000 patients globally". This section effectively explains the complexities of hemostasis in cirrhotic patients. However:
1) The section "Platelets and Bleeding" delves into overly detailed descriptions of platelet physiology. These could be simplified to avoid diverting attention from the primary results and clinical implications. Suggested changes applied.
2) The limitations are candidly discussed, but the strengths could be more prominently emphasized. Suggested changes applied.
3) Table 4 is clear and detailed.
4) Avoid repetition and simplify verbose sections to prioritize key takeaways. Suggested changes applied.
-In the conclusions highlight the practical implications and potential benefits of adopting the new platelet cut-off more clearly. Suggested changes applied.
I suggest this article of which I am co-author to expand line 33-38; I think it is interesting to do a framework on mortality and in what mortality class the disease under study is identified.: Golinelli D, Guarducci G, Sanna A, Lenzi J, Sanmarchi F, Fantini MP, Montomoli E, Nante N. Regional and sex inequalities of avoidable mortality in Italy: A time trend analysis. Public Health Pract (Oxf). 2023 Oct 26;6:100449. doi: 10.1016/j.puhip.2023.100449. Thank you for providing the article.
Reviewer 3 Report
Comments and Suggestions for Authors
The paper by Amin et al is a retrospective study addressing short term re-bleeding and mortality rates of cirrhotic patients undergoing elective esophageal ligation for varices. They used data from a large international database and compared a group of low platelet (between 30 and 49 x103 μ/L) with a group of patients with ≥50 x103 μ/L platelets. Propensity score matching was used and several comorbidity factors were taken into account. The authors concluded that there was no difference in re-bleeding and 14-day mortality between the two groups.
This is an interesting study but there are certain drawbacks due to the nature of the investigation. The authors themselves accept some of them due to the lack of information from the database. The most important are the inability to calculate the MELD score or to evaluate the expertise of the endoscopist. However, there are some additional information that is missing and should be addressed:
1) In the PSM, which factors were taken into account? Factors such as the etiology of cirrhosis, the classification according to Child-Pugh and the presence of ascites and spontaneous bacterial peritonitis may seriously influence the outcome and compromise the conclusions. Most importantly, the presence or absence of any infection is also a critical factor.
2) It is not clear to me why these values of platelets were chosen. In clinical practice, a platelet value below 20x103 is considered serious thrombocytopenia. Therefore, a value between 20 and 30x103 would be more reasonable. What difference one can expect between a patient with 48x103 platelets (first group), and another patient with 51x103 platelets (second group)?
Author Response
The paper by Amin et al is a retrospective study addressing short term re-bleeding and mortality rates of cirrhotic patients undergoing elective esophageal ligation for varices. They used data from a large international database and compared a group of low platelet (between 30 and 49 x103 μ/L) with a group of patients with ≥50 x103 μ/L platelets. Propensity score matching was used and several comorbidity factors were taken into account. The authors concluded that there was no difference in re-bleeding and 14-day mortality between the two groups.
This is an interesting study but there are certain drawbacks due to the nature of the investigation. The authors themselves accept some of them due to the lack of information from the database. The most important are the inability to calculate the MELD score or to evaluate the expertise of the endoscopist. However, there are some additional information that is missing and should be addressed:
1) In the PSM, which factors were taken into account? Factors such as the etiology of cirrhosis, the classification according to Child-Pugh and the presence of ascites and spontaneous bacterial peritonitis may seriously influence the outcome and compromise the conclusions. Most importantly, the presence or absence of any infection is also a critical factor. Thank you for the valuable suggestions. We have rerun the study incorporating the recommended points, including ascites, SBP, PPI use, and variations of the Child-Pugh and MELD scores in the PSM. A detailed list of the covariates before and after PSM is presented in Table 1 of the paper
2) It is not clear to me why these values of platelets were chosen. In clinical practice, a platelet value below 20x103 is considered serious thrombocytopenia. Therefore, a value between 20 and 30x103 would be more reasonable. What difference one can expect between a patient with 48x103 platelets (first group), and another patient with 51x103 platelets (second group)? We chose the value of 30×10³ for two reasons. Firstly, platelets <30×10³ have been shown to be independent predictors of new-onset major bleeding in cirrhotic patients (Drolz et al., https://pubmed.ncbi.nlm.nih.gov/27124745/). Secondly, this threshold is used at our institution by some gastroenterologists for platelet transfusion. We reran the study and performed a subgroup analysis using a threshold of 20×10³, and the findings are presented in the paper. Further studies could evaluate lower thresholds. Lastly, you are correct that the difference between 48×10³ and 51×10³ likely does not have a significant impact on the outcome, but we needed to establish a specific cutoff for the study.
Round 2
Reviewer 2 Report
Comments and Suggestions for Authors
Dear Authors,
The changes have been made well, but the citation of the article in the line 33-38 is missing. The article is: Golinelli D, Guarducci G, Sanna A, Lenzi J, Sanmarchi F, Fantini MP, Montomoli E, Nante N. Regional and sex inequalities of avoidable mortality in Italy: A time trend analysis. Public Health Pract (Oxf). 2023 Oct 26;6:100449. doi: 10.1016/j.puhip.2023.100449.
Author Response
The changes have been made well, but the citation of the article in the line 33-38 is missing. The article is: Golinelli D, Guarducci G, Sanna A, Lenzi J, Sanmarchi F, Fantini MP, Montomoli E, Nante N. Regional and sex inequalities of avoidable mortality in Italy: A time trend analysis. Public Health Pract (Oxf). 2023 Oct 26;6:100449. doi: 10.1016/j.puhip.2023.100449.
We appreciate your comment. The suggested reference/citation is being added. Reference number 2. Thank you!
Reviewer 3 Report
Comments and Suggestions for Authors
The authors have satisfactorily addressed the points raied in the previous review. There are some concerns though on the choice of the range of platelets in the thrombocytopenia group, but this is due to the initial design the source of clinical information. On the other hand, the large number of patients is the strong point of the study. Attention should be made on the confusing overhead of table 2, where both table 1 and table 2 are used at the same table.
Author Response
The authors have satisfactorily addressed the points raied in the previous review. There are some concerns though on the choice of the range of platelets in the thrombocytopenia group, but this is due to the initial design the source of clinical information. On the other hand, the large number of patients is the strong point of the study. Attention should be made on the confusing overhead of table 2, where both table 1 and table 2 are used at the same table.
Thank you for your comment and, more importantly, for bringing the table titles issue to our attention. The headers have been corrected. Thank you again!